# A First-Principles Study on the Electronic, Thermodynamic and Dielectric Properties of Monolayer Ca(OH)_2_ and Mg(OH)_2_

**DOI:** 10.3390/nano12101774

**Published:** 2022-05-23

**Authors:** Mehrdad Rostami Osanloo, Kolade A. Oyekan, William G. Vandenberghe

**Affiliations:** 1Department of Physics, University of Texas at Dallas, Richardson, TX 75080, USA; mxr180067@utdallas.edu; 2Department of Materials Science and Engineering, University of Texas at Dallas, Richardson, TX 75080, USA; kolade.oyekan@utdallas.edu

**Keywords:** 2D dielectric materials, 2D van der Waals dielectrics, 2D dielectrics with TMD channels, 2D heterostructures for FETs, (OH)_2_/HfS_2_ heterobilayer, Mg(OH)_2_/W_2_ heterobilayer

## Abstract

We perform first-principles calculations to explore the electronic, thermodynamic and dielectric properties of two-dimensional (2D) layered, alkaline-earth hydroxides Ca(OH)${}_{2}$ and Mg(OH)${}_{2}$. We calculate the lattice parameters, exfoliation energies and phonon spectra of monolayers and also investigate the thermal properties of these monolayers, such as the Helmholtz free energy, heat capacity at constant volume and entropy as a function of temperature. We employ Density Functional Perturbation Theory (DFPT) to calculate the in-plane and out-of-plane static dielectric constant of the bulk and monolayer samples. We compute the bandgap and electron affinity values using the HSE06 functional and estimate the leakage current density of transistors with monolayer Ca(OH)${}_{2}$ and Mg(OH)${}_{2}$ as dielectrics when combined with HfS${}_{2}$ and WS${}_{2}$, respectively. Our results show that bilayer Mg(OH)${}_{2}$ (EOT∼0.60 nm) with a lower solubility in water offers higher out-of-plane dielectric constants and lower leakage currents than does bilayer Ca(OH)${}_{2}$ (EOT∼0.56 nm). Additionally, the out-of-plane dielectric constant, leakage current and EOT of Mg(OH)${}_{2}$ outperform bilayer h-BN. We verify the applicability of Anderson’s rule and conclude that bilayers of Ca(OH)${}_{2}$ and Mg(OH)${}_{2}$, respectively, paired with lattice-matched monolayer HfS${}_{2}$ and WS${}_{2}$, are effective structural combinations that could lead to the development of innovative multi-functional Field Effect Transistors (FETs).

## 1. Introduction

The isolation of graphene has generated immense attention due to its unique physical properties, such as efficient heat transport, exceptional optical nature (only absorbs 2.3% of light over a wide range of frequencies), ballistic conductance and unprecedented mechanical strength [1,2,3,4]. These properties have motivated many scientists to devote their efforts to exploring other two-dimensional (2D) van der Waals (vdW) materials.

Different experimental techniques, such as mechanical exfoliation [5], epitaxial growth [6] and chemical vapor deposition [7,8] have been employed to synthesize other prominent 2D vdW compounds, such as large and narrow bandgap semiconductors: hexagonal boron nitride (h-BN) [9] and black phosphorus [10,11]. The continuous search for other promising 2D materials led to the discovery of the most recognized family of 2D layered materials—transitional metal dichalcogenides (TMDs)—and prompted investigations into their unique and exceptional electronic, optical and magnetic characteristics [12,13,14,15,16,17,18,19].

Hitherto, most of the 2D vdW materials have been used as channel materials in electronic applications in conjunction with three-dimensional (3D) dielectric materials to create high-performance metal-oxide semiconductor field effect transistors (MOSFETs) [20,21,22]. According to Moore’s Law, increasing circuit complexity through scaling alone is insufficient [23,24]. Instead, the electrical performance of scaled MOSFETs must be enhanced by incorporating low-dimensional materials, such as 2D layered dielectrics [20,25,26].

Although traditional non-vdW dielectrics, such as SiO${}_{2}$ and HfO${}_{2}$, provide high-k solutions for silicon-based semiconductor technologies [27,28,29], they cannot be scaled when grown on 2D channel materials [30,31]. Moreover, the oxide dangling bonds at the interface between a 2D vdW material and the 3D oxides cause an inevitable charge exchange between the oxide and the channel [32,33]. The electric fields resulting from the dangling bonds and the defects introduced in the 2D vdW material will inevitably increase the scattering rate and lower the mobility of carriers in the channel [32,34].

Recently, the scientific community has begun to examine alternate 2D vdW dielectrics to address the problem of unpassivated bonds at the surface observed in HfS${}_{2}$. 2D vdW dielectrics promise a dielectric that is layered and has naturally passivated bonds [35,36]. Integrating vdW channel materials with 2D vdW dielectrics will enable low Equivalent Oxide Thickness (EOT) low defect dielectrics on top of monolayer channels to reach the transistor’s ultimate scaling limit.

Unfortunately, only one van der Waals gate dielectric is currently available: h-BN [37,38]. However, h-BN has an unacceptable leakage current if used in complementary metal-oxide-semiconductor (CMOS) transistors, and its small dielectric constant leads to an unacceptably low capacitive coupling for thicker h-BN layers [36]. Therefore, it is imperative to investigate new classes of layered dielectrics to harness all the possibilities for making a new generation of miniaturized and high-performance transistors (Figure 1).

In our recent works [35,36], we employed Density Functional Theory (DFT) to investigate novel 2D vdW dielectrics that aim to identify new vdW dielectrics. In the first work, we introduced six new candidate materials, namely HoOI, LaOBr, LaOCl, LaOI, $SrI_{2}$ and YOBr as potential candidates for *n*-MOS and *p*-MOS technologies [35].

In a second and more recent work [36], we investigated the dielectric performance of a promising class of Transitional Metal Nitride Halides (TMNHs) to examine their potential applications in *p*-MOS technology when combined with TMD channels. In this second work, a $MoSe_{2}$ channel with HfNCl as a gate dielectric was predicted to be the best combination for a *p*-MOS transistor. Furthermore, there have also been experimental efforts attempting to resolve the issue, as recently reported in Nature Electronics regarding CaF${}_{2}$ and Bi${}_{2}$SeO${}_{5}$ [39,40,41].

CaF${}_{2}$ is a crystalline compound that can potentially address the low dielectric constant (∼3.9), excessive leakage current, premature dielectric breakdown and synthesis high temperature requirement in h-BN [42]. Moreover, CaF${}_{2}$ eliminates some of the drawbacks of an amorphous oxide, such as HfO${}_{2}$ and SiO${}_{2}$. However CaF${}_{2}$ does not alleviate the issue of unpassivated bonds at the surface. Bi${}_{2}$SeO${}_{5}$ is a layered material that was grown as a native oxide on the layered Bi${}_{2}$O${}_{2}$Se [41]; however, it remains unclear how it would be grown on other materials. Consequently, more research needs to be conducted, and delicate experimental methods should be employed to measure and verify the actual performance of the 2D vdW materials and their optical and dielectric properties [43,44,45].

Calcium hydroxide, Ca(OH)${}_{2}$, and magnesium hydroxide, Mg(OH)${}_{2}$, are prominent members of a class of multi-functional 2D layered inorganic compounds with a wide range of applications in cutting-edge technologies in electronics and photo-electronic devices [46,47,48,49]. They are the simplified examples of isomorphous hydroxides with a chemical formula of M(OH)${}_{2}$, where M (=Ca, Mg) is an alkali-earth metal and (OH) is known as hydroxide. The tightly bonded hydrogen and oxygen atoms in -OH groups form chemically passivated surfaces, which explains the stability of these 2D structures under ambient conditions. The molecules in both compounds are held together via ionic bonds between the calcium ion (Ca${}^{2+}$) and two hydroxide ions (OH${}^{-}$) [50,51,52]. We dismissed Ca(OH)${}_{2}$ and Mg(OH)${}_{2}$ in our prior investigation [35] because they are bases, and we assumed they were too soluble in water for practical applications. However, we discovered that the solubility of these materials is not that high and drops dramatically with temperature [53], making them suitable for a wide range of industrial applications. Interestingly, large sample sizes of Ca(OH)${}_{2}$ (>5 mm) and Mg(OH)${}_{2}$ (>8 mm) were recently grown (by 2D semiconductors USA) [54] using the float zone synthesis technique to yield perfectly layered and highly crystalline vdW crystals.

Ca(OH)${}_{2}$, is the only known hydroxide of calcium. The mineral form of Ca(OH)${}_{2}$ is sometimes called portlandite. It is well-established that crystallized Ca(OH)${}_{2}$ in dry air is stable and has an easily cleavable layered brucite type structure [55]. Mg(OH)${}_{2}$ is also found in a mineral form known as abrucite or texalite and can be synthesized using different techniques [55,56]. In addition to their dielectric properties, Ca(OH)${}_{2}$ and Mg(OH)${}_{2}$ are examples of advanced materials with applications in carbon capture and heat storage [57]. Therefore, these materials are available for experimental studies and opportunities for their application in cutting-edge technologies are unexplored.

In this work, we perform accurate first-principles calculations to study the electronic, thermodynamic and dielectric properties of two novel vdW materials: Ca(OH)${}_{2}$ and Mg(OH)${}_{2}$. In addition to the electronic properties of monolayer Ca(OH)${}_{2}$ and Mg(OH)${}_{2}$, we report the exfoliation energies of these two promising layered dielectrics along with their structural stability from their phonon spectrum. We also study their thermodynamic properties, including the free energy, their heat capacitance at constant volume and entropy change at various temperatures.

Moreover, we accurately calculate the macroscopic in-plane and out-of-plane dielectric constants of the bulk and monolayer using Density Functional Perturbation Theory (DFPT). We also use the HSE06 hybrid functional to calculate the bandgaps, electron affinities and the effective masses of charge carriers in the monolayer. We model the performance of each of these materials as a gate dielectric, considering the equivalent oxide thickness (EOT) as well as the leakage current. We consider the performance of monolayer Ca(OH)${}_{2}$ and Mg(OH)${}_{2}$ as dielectrics when combined with monolayer HfS${}_{2}$ and WS${}_{2}$ channels.

Although a stringent lattice matching requirement is not required for 2D heterostructures, the existence of substantial lattice mismatch combined with weak vdW bonding between the 2D layers can result in incoherent lattice matching and the formation of Moiré patterns [58]. Therefore, to design closely aligned heterostructure with minimum lattice mismatch (<0.5%), we integrate Ca(OH)${}_{2}$ with HfS${}_{2}$ and Mg(OH)${}_{2}$ with WS${}_{2}$.

## 2. Computational Methods

We employ DFT as implemented in the Vienna ab initio simulation package (VASP) [59] and use the generalized gradient approximation (GGA) as proposed by Pedrow–Burke–Ernzerof (PBE) for the electron exchange and correlation functional [60]. We set the plane-wave basis energy cut-off to 520 eV for monolayer and bulk Ca(OH)${}_{2}$ and Mg(OH)${}_{2}$. The structural relaxations are continued until the force on each atom is less than $10^{-3}$ eV/Å. For precise phonon and dielectric calculations, we set a tight energy convergence criterion of $10^{-8}$ eV.

We employ the same criteria for monolayer TMDs (HfS${}_{2}$ and WS${}_{2}$) as we used in our prior work in Ref [36]. To mesh the Brillouin Zone (BZ), 12×12×12 and 12×12×1
*k*-points grids are employed for the bulk and the monolayer structures, respectively. For the heterostructures, we used 15×15×1
*k*-points grids for the structural relaxation and 6×6×1 for the HSE06 bandgap calculation. The DFT-D3 approach of Grimme is used to account for interlayer van der Waals interactions [61].

We establish at least a 15 Å vacuum between the monolayers to avoid any non-physical interactions between layers. Moreover, the exfoliation energy is calculated as the ratio of the difference in bulk and monolayer ground state energies to the surface area of the bulks [62,63]. We used the sumo code to plot band structures and Density of States (DOS) [64].

To calculate the phonon spectrum, we used the open-source package; Phonopy [65]. The first-principles phonon calculations with the finite displacement method (FDM) were generated for a set of displacements [4]. We used a 4×4×1 supercell yielding a total of 80 atoms where the atomic displacement distance was $10^{-3}$ Å. When the phonon frequencies over the Brillouin zone were computed, the Helmholtz free energy of the phonons under the harmonic approximation was estimated using the canonical distribution in statistical mechanics as detailed in Refs. [65,66]. Once the phonon frequencies were calculated, we used the thermodynamic relations to determine the thermodynamic properties of the system:(1)A(T)=U(T)−TS(T)
(2)$$C_{v}={\left(\frac{\partial U}{\partial T}\right)}_{V}$$
(3)$$S=\frac{\partial A}{\partial T}$$
where A(T), $C_{v}$ and *S* are the Helmholtz free energy, constant volume heat capacity and entropy, respectively. U(T) = $U_{L}$ + $U_{V}\left(T\right)$ is the phonon-contribution to the internal energy of the system, which is a combination of the lattice internal energy ($U_{L}$) and vibrational internal energy ($U_{L}$).

We calculate the bulk dielectric constants using Density Functional Perturbation Theory (DFPT) as implemented in VASP. We first calculate the permittivity tensor of the bulk unit cell. We then obtain the in-plane and the out-of-plane dielectric constants from the permittivity tensor. We compute the static dielectric constant ($\epsilon_{0}$), which includes both the electronic and ionic responses.

We also determine the optical dielectric constant ($\epsilon_{\infty }$) at high frequency, when only electrons respond to the external field while ions stay fixed in their lattice sites. To acquire the contribution of a monolayer itself, we subtract the vacuum contribution from the supercell dielectric constants and rescale the supercell dielectric constants using the following rescaling formulas [35,38]:(4)$$\epsilon_{2D,⊥}={\left[1+\frac{c}{t}\left(\frac{1}{\epsilon_{SC,⊥}}-1\right)\right]}^{-1}\;$$
(5)$$\epsilon_{2D,\|}=\left[1+\frac{c}{t}\left(\epsilon_{SC,\|}-1\right)\right]\;$$
where *c* and *t* are the size of supercell and the monolayer thickness, respectively.

To calculate the EOT, we use:(6)$$EOT=(\frac{\epsilon_{{\mathrm{SiO}}_{2}}}{\epsilon_{{M(\mathrm{OH})}_{2}}})t$$
where $\epsilon_{{\mathrm{SiO}}_{2}}$, $\epsilon_{{M(\mathrm{OH})}_{2}}$ and *t* are, respectively, the dielectric constant of SiO${}_{2}$ (3.9), the dielectric constant of monolayer M(OH)${}_{2}$ and the thickness of a monolayer M(OH)${}_{2}$.

To compute the thermionic and tunneling current densities through the metal-semiconductor, we use the direct tunneling equations [67,68]:(7)$$J_{\mathrm{tun}}=\frac{q^{3}\;E^{2}}{8\pi h(\varphi -\varphi_{0})}\exp\left(\frac{-4\sqrt{2m^{*}}\;{(\varphi }^{3/2}-\varphi_{0}^{3/2})}{3q\hbar E}\right)\;$$
(8)$$J_{\mathrm{therm}}=A^{**}\;T^{2}\;\exp\left(\frac{-q\left(\varphi -\sqrt{\frac{qE}{4\pi \varepsilon_{i}}}\right)}{kT}\right)\;$$
where E, φ, $\varepsilon_{i}$, $A^{**}$, *q*, $m^{*}$, *k* and *T* are the electric field in the insulator, barrier height, insulator permittivity, effective Richardson constant, electron charge, electron effective mass, Boltzmann constant and temperature, respectively. In Equations (Equation 7) and (Equation 8) φ and $\varphi_{0}$ are the height of energy barrier so that $\varphi_{0}$ = $\varphi -V_{g}$, where $V_{g}$ = $V_{\mathrm{DD}}-V_{\mathrm{thr}}$. $V_{\mathrm{DD}}$ and $V_{\mathrm{thr}}$ are sequentially the supply voltage (0.7 V) and threshold voltage (0.345 V) as determined by IRDS in 2020 [69]. To calculate the electric field inside the insulator, we have:(9)$$E=\frac{\left(V_{\mathrm{DD}}-V_{\mathrm{thr}}\right)}{t}$$
where *t* is the thickness of the monolayer as shown in Figure 2.

We also calculate the electron effective mass in the out-of-plane direction. We derive the effective mass from the curvature of the bulk band structure by considering a 100 *k*-point path along the high symmetry path in out-of-plane direction using the PBE functional:(10)$${\left(m^{*}\right)}^{-1}=\frac{1}{\hbar^{2}}\frac{d^{2}E}{dk^{2}}$$
where E(k) is the energy of the carrier, *k* is the component of the wavevector in the out-of-plane direction, and *ℏ* is the reduced Plank constant. The electron effective mass of Ca(OH)${}_{2}$ and Mg(OH)${}_{2}$ (out-of-plane tunneling mass) are computed to be 0.538 and 0.501, respectively. The effective mass of h-BN (0.47 $m_{e}$) is obtained from Ref. [70].

## 3. Results and Discussion

### 3.1. Structure of Monolayer M(OH)${}_{2}$

Figure 2 depicts side and top views of the atomic structure of the monolayer metal hydroxides under investigation. The layered crystal structure of M(OH)${}_{2}$ has a hexagonal shape with an A-A stacking configuration in which the hydroxides form a relatively close-packed array, marginally expanded in the out-of-plane direction, surrounding cations ${\left(X\right)}^{2+}$ in octahedral coordination. Table 1 reports the structural parameters, such as the monolayer thickness (*t*), bulk interlayer distance (*d*) and the lattice constant of these metal hydroxides.

The monolayer thicknesses (obtained from bilayer) of Ca(OH)${}_{2}$ and Mg(OH)${}_{2}$ are 4.78 and 4.60 Å, whereas the calculated interlayer distance (obtained from bulk) of these compounds is 4.75 and 4.56 Å, respectively. Sequentially, the optimized in-plane lattice constants of monolayer Ca(OH)${}_{2}$ and Mg(OH)${}_{2}$ are calculated to be 3.61 and 3.28 Å. As depicted in Figure 2, the bond lengths of Ca-O (Mg-O) and O-H bonds are calculated to be $l_{1}$ = 2.38 (2.15 Å) and 0.97 Å, respectively. Furthermore, the calculated angles of O-Ca-O (O-Mg-O) and Ca-O-H (Mg-O-H) in the optimized structures are α = 81.13${}^{\circ }$ (80.34${}^{\circ }$) and β = 118.69${}^{\circ }$ (118.07${}^{\circ }$), respectively. The calculated lattice parameters and bond lengths calculated in this study are in consonance with the values reported in previous experimental and theoretical works [50,71,72,73].

We compute the exfoliation energies, E${}_{\mathrm{ex}}$, to verify if Ca(OH)${}_{2}$ and Mg(OH)${}_{2}$ are indeed layered. The calculated exfoliation energy values for Ca(OH)${}_{2}$ and Mg(OH)${}_{2}$ are 33.48 meV/Å^2^ and 57.91 meV/Å^2^, respectively. Taking the general-guideline in Ref. [74] stating that exfoliable compounds have E${}_{\mathrm{ex}}$ /< 100 meV/${Å}^{2}$, both Ca(OH)${}_{2}$ and Mg(OH)${}_{2}$ are easily exfoliable, albeit not as easily exfoliable as TMD materials, which have a E${}_{\mathrm{ex}}$ <25 meV/Å^2^. Experimentally, monolayer Ca(OH)${}_{2}$ and Mg(OH)${}_{2}$ have been exfoliated from the bulk portlandite crystal and from its synthesized bulk crystals onto piranha cleaned 285 nm thermal SiO${}_{2}$/Si substrates [48,75].

### 3.2. Structural Stability and Thermodynamic Properties

Figure 3 shows the 2D phonon dispersion curves for monolayer Ca(OH)${}_{2}$ (Figure 3a) and Mg(OH)${}_{2}$ (Figure 3b). Since the unit cell of monolayer Ca(OH)${}_{2}$ and Mg(OH)${}_{2}$ includes five atoms, the phonon dispersion has twelve optical and three acoustic modes. The phonon spectra, in Figure 3, with strictly positive frequencies clearly indicate that the monolayer of both materials is predicted to be dynamically stable. The E${}_{g}$ and A${}_{1g}$ modes reflect translational motion, implying that the O-H bond distance is maintained for these modes.

However, the E${}_{g}^{\left(\mathrm{OH}\right)}$ and A${}_{1g}^{\left(\mathrm{OH}\right)}$ modes represent the reciprocating vibration of O and H atoms, implying that the O-H bond distance varies. In both compounds, the O-H stretching [50,76] at high frequency mode with energy 470 meV appears in the phonon spectrum. Inspecting the displacement vectors of this high-frequency mode (∼470 meV), we see that it is associated with out-of-plane displacement of the hydrogen and oxygen atoms.

Figure 4 shows the thermodynamic properties of monolayer Ca(OH)${}_{2}$ and Mg(OH)${}_{2}$, in which $S_{v}\left(T\right)$, $C_{v}$ and A(T) are the entropy, the heat capacity at constant volume and the Helmholtz free energy of a 2D system (see computational methods). Our calculation reveals that monolayer Mg(OH)${}_{2}$ has a larger free energy than monolayer Ca(OH)${}_{2}$, while this alone would imply that monolayer Mg(OH)${}_{2}$ is more reactive than monolayer Ca(OH)${}_{2}$, we are only considering the phonon contribution to the internal energy.

Experimental results show that bulk Ca(OH)${}_{2}$ is more reactive than Mg(OH)${}_{2}$ when exposed to carbon dioxide ($CO_{2}$) and this likely remains true in monolayers [77,78]. We observe that $C_{v}$ of monolayer Ca(OH)${}_{2}$ at around room temperature is slightly higher than that of monolayer Mg(OH)${}_{2}$, indicating a greater change in the internal energy in a wide range of temperatures.

The Helmholtz free energy of monolayer Ca(OH)${}_{2}$ is smaller than the Helmholtz free energy of monolayer Mg(OH)${}_{2}$ predicting that the surface structure of Ca(OH)${}_{2}$ over a wide temperature range (from 0 to 1000 K) is more thermodynamically favorable. Thus, from a thermodynamic perspective, monolayer Ca(OH)${}_{2}$ is predicted to be the better material compared to monolayer Mg(OH)${}_{2}$.

### 3.3. Band Structures and DOS

Figure 5 illustrates the band structures and total DOS of monolayer Ca(OH)${}_{2}$ and Mg(OH)${}_{2}$ along the high symmetry *k*-points (Γ-M-K-Γ). We calculated the bandgap of these materials using PBE and HSE06. The HSE06 functional compensates for the bandgap underestimation observed in non-hybrid PBE calculations. The calculated bandgaps from PBE for the monolayer Ca(OH)${}_{2}$ and Mg(OH)${}_{2}$ are 3.68 and 3.42 eV, respectively, whereas the monolayer bandgaps calculated using HSE06 are 5.19 and 4.93 eV. As we see from the band structure, both monolayer compounds have a direct bandgap with valence and conduction bands located at the Γ point. Our calculation predicts that both PBE/HSE06 will have a slightly larger bandgap for monolayer Ca(OH)${}_{2}$ compared to Mg(OH)${}_{2}$.

To explore the insulating properties of Ca(OH)${}_{2}$ and Mg(OH)${}_{2}$ monolayer dielectrics, we compute the band offset of monolayer Ca(OH)${}_{2}$ and Mg(OH)${}_{2}$ and compare it with two different TMD channels: HfS${}_{2}$ and WS${}_{2}$. In addition to exfoliability and stability as characteristics of a good vdW layered dielectric, a desirable dielectric must be a good insulator with a suitable dielectric-channel band offset surpassing at least 1 eV to reduce the leakage current through tunneling or thermionic emission.

Figure 6a displays the electron affinity, the bandgap and the relative position of the band edges (with respect to the vacuum level) of monolayer Ca(OH)${}_{2}$ and Mg(OH)${}_{2}$ with HfS${}_{2}$ and WS${}_{2}$ as channel materials. The conduction band and the valence band edges are indicated by the solid green and yellow lines, respectively. As shown in Figure 6a, the 1 eV band offset requirement of each dielectric with the valence band of the proposed TMD channels (HfS${}_{2}$ and WS${}_{2}$) is not met, signifying that only designing an *n*-MOS transistor with Ca(OH)${}_{2}$/HfS${}_{2}$ and Mg(OH)${}_{2}$/WS${}_{2}$ is theoretically feasible.

Therefore, we only evaluate the performance of these dielectrics in an *n*-MOS transistor. Figure 5b displays the averaged potential of the heterostructures (Ca(OH)${}_{2}$/HfS${}_{2}$ and Mg(OH)${}_{2}$/WS${}_{2}$) with respect to the distance along the z-axis of the supercell lattice. Applying Anderson’s rule, the HSE bandgaps for Ca(OH)${}_{2}$/HfS${}_{2}$ and Mg(OH)${}_{2}$/WS${}_{2}$ are calculated to be 1.20 and 1.99 eV, while we calculate 0.97 and 2.11 eV band offset for monolayer Ca(OH)${}_{2}$ and Mg(OH)${}_{2}$ when combined with HfS${}_{2}$ and WS${}_{2}$, respectively.

As depicted, the electron affinity(χ), is the difference between the vacuum level and the Fermi level (vacuum level is shifted to zero). The electron affinity of Ca(OH)${}_{2}$/HfS${}_{2}$ and Mg(OH)${}_{2}$/WS${}_{2}$ obtained from HSE06 are calculated to be 3.12 and 3.00 eV, respectively. We also include the band edges of each M(OH)${}_{2}$ and TMD compare with Anderson’s rule [79,80] for heterostructures.

As depicted in Figure 6b, the predicted HSE bandgap using Anderson’s rule for the Ca(OH)${}_{2}$/HfS${}_{2}$ and Mg(OH)${}_{2}$/WS${}_{2}$ heterostructures are in good agreement with the calculated HSE bandgap for Mg(OH)${}_{2}$ WS${}_{2}$ channel.

Table 2 shows the calculated bulk dielectric constants in the in-plane (‖) and out-of-plane (⊥) directions. In order, the calculated in-plane static dielectric constants of Ca(OH)${}_{2}$ and Mg(OH)${}_{2}$ are 12.30 and 9.75 while in the out-of-plane direction, the calculated values are 4.53 and 4.32, respectively. In both materials, the optical dielectric constant is much lower in the in-plane direction than in the out-of-plane direction, similar to what we found in previous studies on layered materials [35,38].

We find that the ionic contribution to the static dielectric constant in the in-plane direction for bulk Ca(OH)${}_{2}$(Mg(OH)${}_{2}$) is large and accounts for 338% (261%), while in the out-of-plane direction, the ionic response contributes 75% (62%) to the static dielectric constant for bulk Ca(OH)${}_{2}$(Mg(OH)${}_{2}$).

To compute the monolayer dielectric constant, we isolate monolayers in a computational supercell with sufficient vacuum, then we rescale the supercell’s estimated dielectric values to the monolayer’s, as detailed in the computational methods. Our calculations demonstrate that monolayer of Ca(OH)${}_{2}$ and Mg(OH)${}_{2}$ have static in-plane dielectric constants of 8.94 and 7.75, whereas their out-of-plane static dielectric constants are 6.40 and 6.33, respectively. Our results show a decrease in the in-plane dielectric constant when we move from bulk to monolayer, while the trend is increasing in the out-of-plane direction.

Compared to the dielectric constants of wide bandgap ionic crystals, such as ZnCl${}_{2}$ (4.00), CaHBr (4.60) and MgF${}_{2}$ (5.40) [35,81]; monolayer Ca(OH)${}_{2}$ offers a higher out-of-plane static dielectric constant. Moreover, the dielectric constants of a monolayer Ca(OH)${}_{2}$ and Mg(OH)${}_{2}$ are greater than the dielectric constant of monolayer h-BN (3.29) [35,38]. Moreover, while Ca(OH)${}_{2}$ and Mg(OH)${}_{2}$ do not match the dielectric constants of “high-k” materials, such as HfO${}_{2}$, it is significantly higher than SiO${}_{2}$ or h-BN [28,82].

### 3.4. Tunneling Current and Dielectric Performance

To design a viable transistor, a dielectric with a small thickness, a high dielectric constant, and a low leakage current is desirable. As outlined in the computational methods, to quantify the promise of a gate dielectric material we compute the leakage current accounting for direct tunneling and thermionic emission for low power devices. To find the best dielectrics, we compute the EOT of Ca(OH)${}_{2}$ and Mg(OH)${}_{2}$ and leakage current, assuming the channel materials (HfS${}_{2}$ and WS${}_{2}$), in an *n*-MOSFET with an electron affinity of 4.98 and 3.73 eV, respectively.

In addition to the stability and lower leakage current criteria, a suitable dielectric candidate should have a small EOT to ensure acceptable electrostatic control. As a reference and for a better comparison of the performance of a device, the leakage current of monolayer h-BN is also calculated. According to the International Roadmap for Devices and Systems (IRDS) [83], the absolute maximum leakage current for any feasible gate dielectric is less than 100 pA/μm per pitch for a transistor with a 28 nm pitch, an effective gate width of 107 nm and a 18 nm long gate.

With these criteria, the acceptable current density is about 0.145 A/cm^2^. We remark that we used a larger *k*-grid (15 × 15 × 1) for the HSE calculations in this study, improving over previous estimations [35,36,38].

Table 3 shows the leakage currents of Ca(OH)${}_{2}$ and Mg(OH)${}_{2}$ and the calculated EOTs. Our calculations explicitly show that a monolayer of Mg(OH)${}_{2}$ with an EOT ≤ 0.3 nm satisfies the minimum leakage current criteria as determined by IRDS, whereas Ca(OH)${}_{2}$ when combined with HfS${}_{2}$ does not adequately block leakage current. In comparison, a very small physical thickness of monolayer h-BN and its small dielectric constant result in high leakage currents, making monolayer h-BN unfit for use as a gate insulator in 2D transistors [84].

We added the leakage current of bilayers for the purpose of comparison. Although monolayer h-BN is not sufficiently insulating for *n*-MOS, bilayer h-BN with a higher physical thickness has a smaller leakage current (<$6.37\times 10^{-13}$ A/cm^2^) acceptable by IRDS. According to Table 3, a monolayer of Ca(OH)_2_ with a HfS_2_ channel is not sufficiently insulating, however, a bilayer of Ca(OH)_2_ still has a low EOT (~0.56 nm) while small leakage current is small (<6.06 × 10^−8^ A/cm^2^). Bilayers have double the physical thickness, dramatically reducing the tunneling probability, which is captured by the appearance of the electric field in the exponential in Equation (Equation 7).

Finally, taking the result from our previous study, monolayer and bilayer of LaOCl considerably outperform both monolayer and bilayer of M(OH)${}_{2}$ and h-BN when combined with HfS${}_{2}$ and WS${}_{2}$ channels. We observe that the best performance of a dielectric/channel heterostructure is found for monolayer LaOCl/HfS${}_{2}$ with a small EOT (0.05 nm) and leakage current ($4.79\times 10^{-21}$ A/cm^2^), while alternative layered dielectrics, such as LaOCl, may give even better performance, Ca(OH)${}_{2}$ and Mg(OH)${}_{2}$ are both found in nature and can be commercially synthesized on sizeable crystals.

There are, however, a few unanswered questions that need further clarification:(i)Solubility is critical factor for different applications. Interestingly, at room temperature the solubility of Mg(OH)${}_{2}$ (∼9.80 × $10^{-4}$ g/mL) is very close to the solubility of amorphous SiO${}_{2}$ (∼1.2 × $10^{-4}$ g/mL) [85,86]. In contrast to SiO${}_{2}$, due to the positive heat of solution in M(OH)${}_{2}$ materials M(OH)${}_{2}$ solubility decreases with increasing temperature [85]. Although at room temperature the solubility of Ca(OH)${}_{2}$ (∼1.09 × $10^{-1}$ g/mL) is significantly higher than the solubility of amorphous SiO${}_{2}$ (∼1.2 × $10^{-4}$ g/mL), the solubility of both in water is the same at around 400 K [85,87]. At higher temperatures Ca(OH)${}_{2}$ has a lower solubility compared to amorphous SiO${}_{2}$. Hence, due to high level of reactivity of M(OH)${}_{2}$ materials with water a proper encapsulation architecture is required to enhance the long-term stability of M(OH)${}_{2}$ based devices.(ii)Although both Ca(OH)${}_{2}$ and Mg(OH)${}_{2}$ are stable against oxidation at room temperature, they are not stable against CO${}_{2}$ [85]. Consequently, these materials have been used for carbon capture and thermal heat storage. M(OH)${}_{2}$ materials and other members of this family (i.e.,Cd(OH)${}_{2}$, Ni(OH)${}_{2}$, Zn(OH)${}_{2}$) have the potential to contribute to the worldwide goal of decarbonization through carbon capture and storage, which would help to protect our planet from the catastrophic effects of climate change [77,88,89].(iii)We combined Ca(OH)${}_{2}$ and Mg(OH)${}_{2}$ with two TMD channels (HfS${}_{2}$ and WS${}_{2}$) in this work; there are a handful of M(OH)${}_{2}$/TMD (i.e., Mg(OH)${}_{2}$/MoS${}_{2}$) combinations that need to be carefully investigated to find the most promising heterostructure candidates for *p*-MOS and *n*-MOS applications. The bandgap tuning of the heterostructures in the presence of an external field [71,72], the effect of strain and stress, as well as the defect formations in monolayers and bilayers, are other interesting topics for future studies.

## 4. Conclusions

We investigated two novel 2D layered materials, Ca(OH)${}_{2}$ and Mg(OH)${}_{2}$, for their potential applications as dielectrics in *n*-MOS devices. For each material, we calculated the exfoliation energy, band offset, phonon spectrum, thermodynamic properties, EOT and leakage current. The exfoliation energies confirmed that both materials are mechanically exfoliable and can be isolated in layers. The strictly positive phonon spectra clearly demonstrated the structural stability of the monolayers. We studied the thermodynamic properties and observed that the smaller free energy of monolayer Ca(OH)${}_{2}$ makes its surface more favorable for applications compared to monolayer Mg(OH)${}_{2}$.

We also used DFPT to calculate the in-plane and out-of-plane macroscopic dielectric constants. Although the in-plane static dielectric constant of monolayer Ca(OH)${}_{2}$ is 15% higher than that of Mg(OH)${}_{2}$, in the out-of-plane direction, a single-layer of Mg(OH)${}_{2}$ had a relatively higher dielectric constant (6.40) compared with Ca(OH)${}_{2}$ (6.33). We calculated the leakage current and the EOT for each material to evaluate their performance as a gate dielectric when combined with TMD channel materials.

Our calculations showed that the bilayer Mg(OH)${}_{2}$/WS${}_{2}$ heterostructure offers a lower leakage current compared to the Ca(OH)${}_{2}$/HfS${}_{2}$ heterostructure, while other layered dielectrics, such as LaOCl, could provide even better performance. Ca(OH)${}_{2}$ and Mg(OH)${}_{2}$ are available both in nature and commercially synthesized on sizeable crystals. Moreover, we validated that the predicted HSE bandgaps using Anderson’s rule for the Ca(OH)${}_{2}$/HfS${}_{2}$ heterostructure were in good agreement with the calculated HSE bandgap for Ca(OH)${}_{2}$ and its associated TMD channel, HfS${}_{2}$.

The band offset results in Ca(OH)${}_{2}$/HfS${}_{2}$ and Mg(OH)${}_{2}$/WS${}_{2}$ heterostructures using Anderson’s rule were 1.20 and 1.99 eV, and we calculated 0.97 and 2.11 eV band offsets for monolayer Ca(OH)${}_{2}$ and Mg(OH)${}_{2}$ when combined with HfS${}_{2}$ and WS${}_{2}$, respectively. The calculated leakage currents for 2L Ca(OH)${}_{2}$ and Mg(OH)${}_{2}$ were much lower than the IRDS requirement, and their 2L EOTs were calculated to be 0.56 and 0.60 nm, respectively.

Our results show that a FET with a monolayer of Mg(OH)${}_{2}$ as a dielectric would outperform monolayer Ca(OH)${}_{2}$ and h-BN. We expect that our findings, along with the recent crystalline synthesis of Ca(OH)${}_{2}$ and Mg(OH)${}_{2}$, will lead to more research regarding the combination of novel 2D layered dielectrics with other prominent TMD channels for applications in 2D FETs.

## Figures and Tables

**Figure 1 nanomaterials-12-01774-f001:**
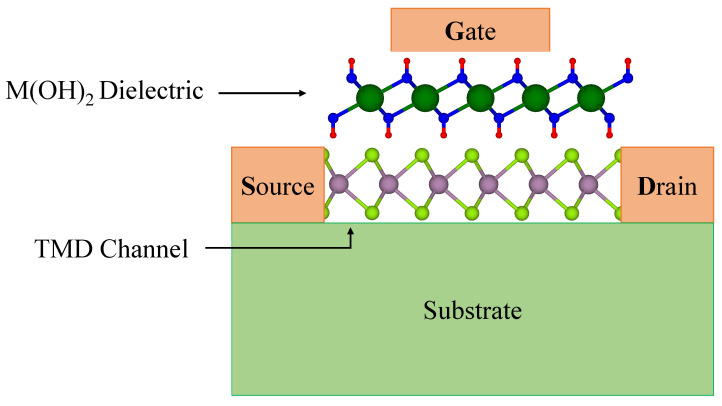
A schematic view of a FET made from a monolayer M(OH)${}_{2}$ dielectric and a monolayer TMD channel.

**Figure 2 nanomaterials-12-01774-f002:**
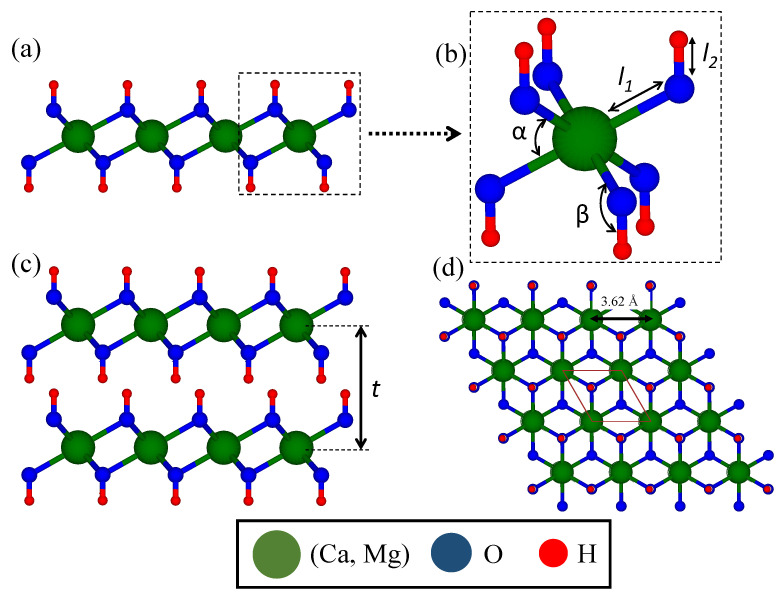
Structures of alkaline-earth metal hydroxides. The side view (**a**) and top view of the monolayer (**d**) in addition to the side view of the bilayer (**c**) are illustrated. (**b**) demonstrates Ca-O/Mg-O ($l_{1}$) and O-H bond length ($l_{2}$) along with the angles between bonds (α and β). The monolayer thickness (*t*) is indicated on the bilayer structures.

**Figure 3 nanomaterials-12-01774-f003:**
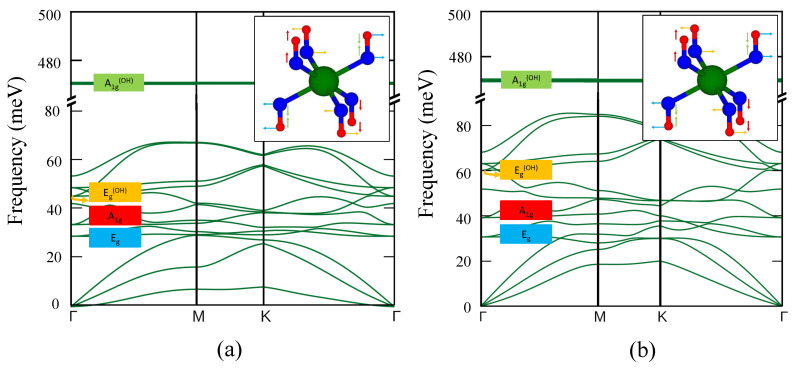
M(OH)${}_{2}$’s phonon dispersion curves. (**a**) Phonon dispersion spectrum of Ca(OH)${}_{2}$. (**b**) Phonon dispersion spectrum of Mg(OH)${}_{2}$. E${}_{g}$ (blue), A${}_{1g}$ (red), E${}_{g}^{\left(\mathrm{OH}\right)}$ (orange) and A${}_{1g}^{\left(\mathrm{OH}\right)}$ (green) modes, respectively, represent the translational motion and the reciprocating motion of the O-H bonds. The insets visualize the in-plane and out-of-plane vibrational modes of O-H. The broken axis represents that there are no phonon branches between 90 and 450 meV. The flat energy curves at high-energy modes (∼470 meV) are associated with out-of-plane hydrogen and oxygen displacements.

**Figure 4 nanomaterials-12-01774-f004:**
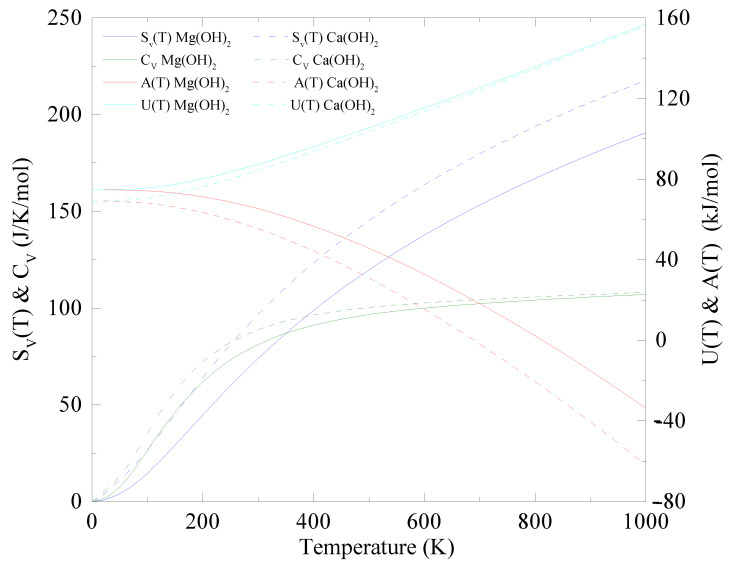
Thermodynamic properties of Ca(OH)${}_{2}$ and Mg(OH)${}_{2}$. Entropy($S_{v}\left(T\right)$), constant volume heat capacity ($C_{v}$), internal energy (*U*(*T*)) and Helmholtz free energy (*A*(*T*)) of Mg(OH)${}_{2}$ (Ca(OH)${}_{2}$) are shown with solid (dashed) blue, green, cyan and red lines, respectively. The smaller Helmholtz free energy implies that from a thermodynamic perspective, monolayer Ca(OH)${}_{2}$ is the better material compared to monolayer Mg(OH)${}_{2}$.

**Figure 5 nanomaterials-12-01774-f005:**
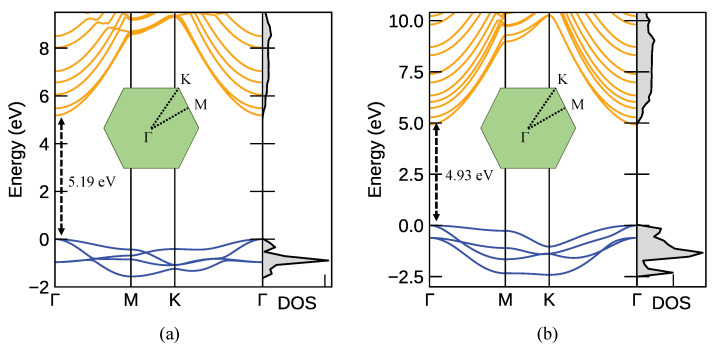
Band structure and total DOS of (**a**) monolayer Ca(OH)${}_{2}$ and (**b**) Mg(OH)${}_{2}$. HSE06 hybrid functionals are used to correct for the usual bandgap underestimation. Both materials have a direct bandgap with the Ca(OH)${}_{2}$ bandgap (5.19 eV) slightly larger than the Mg(OH)${}_{2}$ bandgap (4.93 eV).

**Figure 6 nanomaterials-12-01774-f006:**
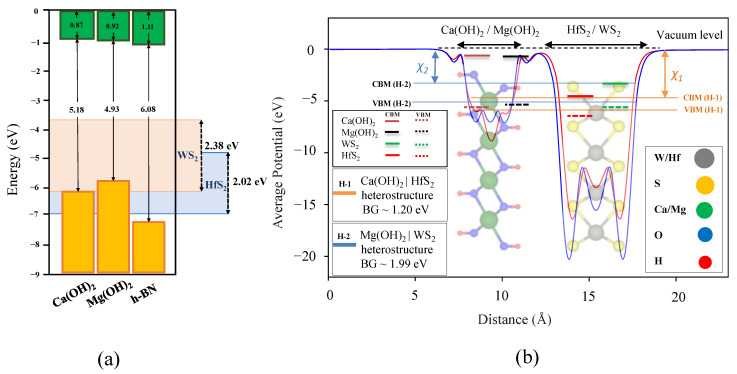
Band alignment of Ca(OH)${}_{2}$ and Mg(OH)${}_{2}$. (**a**) The band offset of HfS${}_{2}$ and WS${}_{2}$ with Ca(OH)${}_{2}$, Mg(OH)${}_{2}$. The vacuum level is set to zero and a monolayer of h-BN is included for the purpose of comparison. (**b**) The average potential of the heterostructures (Ca(OH)${}_{2}$/HfS${}_{2}$ and Mg(OH)${}_{2}$/WS${}_{2}$) is shown with respect to the z-direction, perpendicular to the plane of sheets. Conduction band maximum (CBM) and valence band minimum (VBM) of each material is demonstrated with colored solid and dashed lines. The CBM and VBM of each heterostructure are shown with long solid horizontal lines. Anderson’s rule is used to calculate the bandgap and the electron affinity of the each heterostructure.

**Table 1 nanomaterials-12-01774-t001:** Structural parameters, monolayer thickness, interlayer distance and exfoliation energies of Ca(OH)${}_{2}$ and Mg(OH)${}_{2}$. Exfoliation energies < 100 meV/Å^2^ indicate easily exfoliable materials [74].

Material	Bulk	Monolayer			Monolayer Supercell Thickness (Å)	Exfoliation Energy (meV/Å^2^)
	di (Å)	a (Å)	t (Å)	Δ (%)		
Ca(OH)${}_{2}$	4.75	3.61	4.78	0.63	24.32	33.48
Mg(OH)${}_{2}$	4.56	3.28	4.60	0.88	29.43	57.91

**Table 2 nanomaterials-12-01774-t002:** Static dielectric constants of bulk and monolayer Ca(OH)${}_{2}$ and Mg(OH)${}_{2}$.

Material	Monolayer ($\varepsilon_{\infty }$)		Monolayer ($\varepsilon_{0}$)		Bulk ($\varepsilon_{\infty }$)		Bulk ($\varepsilon_{0}$)	
	//	⊥	//	⊥	//	⊥	//	⊥
Ca(OH)${}_{2}$	2.72	3.18	8.94	6.33	2.81	2.59	12.30	4.53
Mg(OH)${}_{2}$	2.68	3.38	7.75	6.40	2.70	2.66	9.75	4.32

**Table 3 nanomaterials-12-01774-t003:** Leakage current density for *n*-MOS applications through a monolayer (1L) and bilayer (2L) Ca(OH)${}_{2}$ and Mg(OH)${}_{2}$ with HfS${}_{2}$ and WS${}_{2}$ as channel materials, respectively. For comparison, the leakage current of monolayer and bilayer h-BN as well as LaOCl are included.

Materials	Out-of-Plane Electron Effective Mass	Leakage Current (A/cm^2^)		Monolayer EOT (nm)
		1L	2L	
Ca(OH)${}_{2}$ & HfS${}_{2}$	0.538	4.38 $\times \;10^{4}$	6.06 $\times \;10^{-8}$	0.28
Mg(OH)${}_{2}$ & WS${}_{2}$	0.501	1.55 $\times \;10^{-1}$	1.40 $\times \;10^{-18}$	0.30
h-BN & HfS${}_{2}$	0.501	8.95 $\times \;10^{1}$	2.30 $\times \;10^{-13}$	0.38
h-BN & WS${}_{2}$	0.501	1.37 $\times \;10^{0}$	6.37 $\times \;10^{-17}$	0.38
LaOCl & HfS${}_{2}$	1.125	4.79 $\times \;10^{-21}$	3.71 $\times \;10^{-45}$	0.05
LaOCl & WS${}_{2}$	1.125	5.98 $\times \;10^{-4}$	3.64 $\times \;10^{-23}$	0.05

## Data Availability

The data that support the findings of this study are available upon reasonable request.

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
