# Peer review of "A First-Principles Study on the Electronic, Thermodynamic and Dielectric Properties of Monolayer Ca(OH)_2_ and Mg(OH)_2"

_nanomaterials, 2022, doi:10.3390/nano12101774_

Round 1

Reviewer 1 Report

The authors have studied the electronic and thermodynamic properties
of Ca(OH)2 and Mg(OH)2 monolayers (and bulk). The authors are particularly interested in whether these materials could be used as gate dielectrics in field-effect transistors involving 2D material channels. I am not particularly familiar with the literature of these materials, but it seems that, while many of the basic properties of have already been calculated, the thermodynamic properties and the analysis in the context of 2D nanoelectronics is novel.

The paper is mostly well written, the calculations expertly carried out and clearly analyzed. Especially, I was happy to see that the dielectric constants of monolayers are properly treated. There were many minor issues in the presentation, which are listed below, but overall I think the manuscript is of sufficiently high quality to warrant publication in Nanomaterials after minor revision.

- EOT abbreviation used several times before defined at the end of Introduction.

- I do not see the advantage of using bilayer to estimate monolayer thickness: (a) the value is very similar to bulk and (b) both will be equally "off" when contacted with channel and gate.

- Line 205: "thermodynamically stable"->"dynamically stable"

- Line 208: "anharmonic O-H stretching", why do you call this anharmonic?

- Figure 2, please explain what is shown in insets.

- Line 214: "Our calculation reveals a steeper change in the entropy for Ca(OH) 2 which is an indicator of a higher rate of chemical reactivity." I do not understand why that is the case or what is the "rate of chemical reactivity". Please add explanation or citation.

- Figure 3: The authors could also add curves of internal energy U(T) to the figure.

- Table 1 caption : "The in-plane dielectric monolayer Ca(OH)2 is higher
constant, while in the out-of-plane direction, Mg(OH)2 has a higher dielectric constant." It is strange to comment this on the table caption as it seems to refer only to eps_infinity of bulk results.

- Line 257: Hf2 -> HfS2

- Figure 5: The labels are not included in the caption. Panel (b) is very busy, and it takes a lot of time to understand what is happening in it. At minimum, describe it properly in the caption.

- Line 266: "ionic contribution to the static dielectric constant accounts for 335% (260%) for the in-plane dielectric response in Ca(OH)2 (Mg(OH)2)." The way percentages are calculated does not match with the way the sentence is constructed.

- Line 296: "We remark that we used a larger k-grid (15 × 15 times 1) for the HSE calculations in this study, , improving over previous estimations [35,36,38]." What is improved, how much, and why? The k-grid should always be benchmarked to yield converged results. Were they not converged in Refs. 34,36,38? Are they now? Please also fix the typos in the sentence.

Author Response

We thank the reviewers for their time and consideration. Please see the attachment (response letter and revised manuscript).

Reviewer 2 Report

In this work, the authors reported some theoretical simulation on the physical properties of monolayer Ca(OH)2 and Mg(OH)2 and their applications in CMOS devices. Considering the description of numerical methods, the simulated results are believed to be correct, and the authors have calculated the tunneling current densities of Ca(OH)2/Mg(OH)2-based CMOS devices, which are useful in this field and the potential applications of these two monolayers. However, the manuscript is not well written, and major revision is required for the possible publication. I have listed my comments as follows,

major comments,

1) The discussion about the solubilities and etc in the conclusion section can be moved to the Results and Discussions section.

2) A schematic view of the CMOS device should be provided in the manuscript.

3) The method used to calculate EOT should be described in the second section.

4) When constructing Ca(OH)2/HfS2 and other n-MOS structures, have the authors relaxed the heterostructures for further HSE calculations using the method such as those described in Phys. Chem. Chem. Phys., 20:30351?

5)From Table 3, it seems that bilayer of Ca(OH)2 or Mg(OH)2 possesses quite different properties from monolayer.  The stacking manner and the corresponding electronic bandstructures should be described.  And the mechanism to determine the difference in leakage tunneling current density of bilayer and monolayer Ca(OH)2 or Mg(OH)2 should be described as well.

minor comments,

1) although the English is well written in this manuscript, there are still some writing errors, which should be corrected, e.g.

On page 3, "While in our previous study [35], we dismissed Ca(OH)2 and Mg(OH)2 because they are elemental bases and soluble in water, they are not very soluble and their solubility decreases significantly with temperature[53] making these compounds suitable for variety of industrial applications."

On page 5, "In Eq. 3 and Eq. 4 φ and φ0 are the height of energy barrier so that ..."

On Page 10, EOT should be defined when it was firstly used.

On Page 10, "15×15 times 1" 

2) A k-mesh of "12×12×1" is generally dense enough for the DFT calculations of two-dimensional materials. Have the authors conducted the convergence test of electron calculations regarding k-mesh?

3) On page 7, can the authors tell why the O-H bonds are anharmonic?

Author Response

(The authors gave the same response as above.)
